# OpenReview forum: "ALMANACS: A Simulatability Benchmark for Language Model Explainability"
_ICLR.cc/2024/Conference — Submitted to ICLR 2024_

### Official Review · Reviewer_vWZJ · 2023-10-29

**Soundness:** 2 fair
**Presentation:** 1 poor
**Contribution:** 2 fair
**Rating:** 3
**Confidence:** 3

**Summary:**

This paper introduces ALMANACS, a new benchmark for evaluating explainability methods for language models. The key ideas are:

-ALMANACS measures the simulatability of explanations, i.e. how well they help predict model behavior on new inputs. This relates to the desired properties of faithfulness and completeness.
-The benchmark comprises 12 topics with safety-relevant scenarios. Questions are non-objective and designed to elicit complex, nonlinear behavior from models.
-There is distributional shift between train and test sets to require generalization.
-The authors test counterfactual, rationalization, and salience-based explanation methods. None consistently improve upon a no-explanation baseline, indicating simulatability on ALMANACS remains an open challenge.

**Strengths:**

(1) Partially addresses the need for standardized benchmarks to evaluate and compare explanation methods.

(2) Simulatability is a useful metric directly related to explanation quality.

(3) Automated evaluation enables efficient benchmarking.

(4) Non-objective questions and distribution shift require explanations to provide true insight rather than leveraging correlations.

**Weaknesses:**

(1) Only safety-relevant scenarios are included and this is not general.

(2) The choice of language models for evaluation versus being explained may affect results. More analysis of this factor could be useful.

(3) Automated evaluation using a language model proxy for humans has limitations vs. human studies. Direct comparisons would be needed to validate the benchmark.

(4) Testing on more model sizes, scaling effects, and model families instead of just focusing on flan-alpaca-gpt4-xl and vicuna-7b-v1.3.

**Questions:**

Missing references for explanation distillation:

(1) Li et al. Explanations from Large Language Models Make Small Reasoners Better. 2022.

---

> ### Author Response · Authors · 2023-11-20
> **Author Response to Reviewer vWZJ**
>
> Thank you for your review. If we understand correctly, your main concerns are that our experiments are not comprehensive enough. In particular, you would like to see tests with more model sizes and model families, more pairs of language models used for prediction with language models being explained, and human studies. While we leave human studies as an important direction for future work, **we have added experiments with 11 additional language models**. These include nine new language models being explained (in Appendix G) and two new language model predictors (in appendix J).
>
> **Do these changes address your concerns about the comprehensiveness of our experiments?** If not, we would be happy to improve the paper based on further input from you.
>
> > Only safety-relevant scenarios are included and this is not general.
>
> Our work contains 12 different scenario topics, and it has a total of 180,000 training examples and 18,000 test examples.
>
> We are confused how this many different scenario topics and dataset examples is not general enough for ICLR. There are other natural language benchmarks published in ICLR that contain fewer topics and dataset examples. For instance:
> - “WikiWhy: Answering and Explaining Cause-and-Effect Questions”. ICLR 2023 (notable top 5%). The dataset has 9,406 “why” question-answer-rationale triplets spanning 11 topics.
> - “ReClor: A Reading Comprehension Dataset Requiring Logical Reasoning”. ICLR 2020. The dataset has 6,138 questions (and no topic categories).
>
> We hope that our work is judged consistently by the same standards as other papers.
>
> > The choice of language models for evaluation versus being explained may affect results.
>
> To test if our choice of GPT-4 for evaluation affected the results, **we ran additional experiments with two additional language models for evaluation. The results are in Appendix J.** We find that no combination of predictor/explanation, for either model being explained, performs significantly better than using GPT-4 for evaluation with no explanations provided, or than the logistic regression baseline. Consistent with our main results, this indicates that none of these predictors are able to extract helpful information from the explanations beyond what is present in the model’s answers.
>
> > Automated evaluation using a language model proxy for humans has limitations vs. human studies.
>
> This is an important point; human studies are a valuable direction for future work. Nevertheless, even if an automated evaluation isn’t totally consistent with a human study, we still think automated evaluation has a valuable role to play. Because automated evaluation is an order of magnitude faster than human evaluation, it can provide quick, coarse-grained feedback throughout the interpretability researcher’s development cycle. A tool like ALMANACS is the difference between waiting weeks or months for the results of a human study, versus getting results of an automated benchmark in just a few hours.
>
> > Testing on more model sizes, scaling effects, and model families instead of just focusing on flan-alpaca-gpt4-xl and vicuna-7b-v1.3.
>
> **We have added scaling experiments to the paper, in Appendix G.** These experiments test models of a **variety of families with sizes ranging from 0.77B to 20B parameters on the advanced_ai_risk section of ALMANACS**. The benchmark difficulty, as measured by how well logistic regression is able to predict model behavior, is shown in Figure 10. We see a general trend that the behavior of larger models is harder to predict. The trend is more pronounced when we compare benchmark difficulty with model performance on a related task, with more capable models having behavior that is more difficult to predict.
>
> > Missing references for explanation distillation:
> >
> > (1) Li et al. Explanations from Large Language Models Make Small Reasoners Better. 2022.
>
> Thank you. We have added this reference to the paper.

---

> > ### Author Response · Authors · 2023-11-22
> >
> > With one day left in the discussion period, we are hoping to hear a reply from you. Please let us know if we have addressed your concerns.

---

### Official Review · Reviewer_3Mpe · 2023-11-01

**Soundness:** 3 good
**Presentation:** 2 fair
**Contribution:** 2 fair
**Rating:** 3
**Confidence:** 3

**Summary:**

This submission introduces ALMANACS, a novel benchmark tailored for evaluating the explainability of language models via a concept termed "simulatability." Using GPT-4 as a predictor, the benchmark assesses how well GPT-4 can simulate other language models that employ various explanation methods. Simulatability is defined by measuring the distribution distance between the outputs of GPT-4 and the target language model for previously unseen test tasks. A noteworthy finding is that the incorporation of explanations does not invariably enhance explanation performance for unseen inputs.

**Strengths:**

The paper innovatively offers a benchmark with a quantitative metric for assessing explainability in language models. Furthermore,it is interesting for the discovery that models with explanation input do not outperform non-explanatio in terms of simulatability.

**Weaknesses:**

The primary focus of this paper appears to be on the introduction of the ALMANACS benchmark. Yet, the utilization of well-established distance measures like KLDiv and TVDist doesn't add a novel dimension to the study.

The observation that explanation techniques might not always heighten performance on unseen data is compelling, but the paper would benefit from a deeper analysis and discussion on the possible reasons behind this phenomenon.

The term "simulatability" appears to be inconsistently defined, leading to confusion. The initial definition of simulatability is "how well the explanations improve behavior prediction on new inputs". Subsequently, it seems to change to a definition centered on distribution distance, KLDiv or TVDist.

The paper doesn't provide a convincing argument for why simulatability is a good metric for language model explainability. Many factors can influence predictor outputs. Given that GPT-4 operates as a black box, it's hard to say GPT-4 predict solely to the presence or absence of explanations without providing additional constraints. This point is underscored by results from the NoExpl vs Expl comparison in Table 1, which indicates low KLDiv scores, hinting that GPT-4's predictions might be independent of input explanations.

**Questions:**

1. Could the authors provide the precise definition of "simulatability"?
1. The choice of GloVe embeddings for demonstration retrieval appears outdated. Have the authors considered more recent sentence embeddings, such as SimCSE?
1. There seems to be a discrepancy between the comparison results for NoExpl in Figure 4 and Table 1 (PredictAverage vs NoExpl). PredictAverage outperforms NoExpl in Figure 4 but does not in Table 1.
Could this be clarified?

---

> ### Author Response · Authors · 2023-11-20
> **Author Response to Reviewer 3Mpe**
>
> > the paper would benefit from a deeper analysis... on the possible reasons [why explanations don't help on unseen data]
>
> Thank you for this excellent suggestion. **In Appendix H, we have added additional analysis and discussion of possible reasons why the explanation methods are not improving performance.** Specifically, we provide examples of the explanation methods from our dataset, and we give a qualitative analysis of how much information is contained in the explanations. One insight is that to us, the authors of the study, the tokens with high salience scores do not intuitively suggest how the model will behave on new inputs. For example, the tokens are sometimes the question words, such as “would” and “choose”, rather than the details of the scenario. Another observation is that the language model’s rationalizations often discuss the specifics of the input scenario without discussing possible variations of the scenario. This might be limiting how much the rationalizations help the predictor at test-time, when the predictor is presented with scenario variations.
>
> > The term "simulatability" appears to be inconsistently defined.... The initial definition of simulatability is "how well the explanations improve behavior prediction on new inputs". Subsequently, it seems to change to a definition centered on distribution distance, KLDiv or TVDist…. Could the authors provide a precise definition of “simulatability”?
>
> It seems our original submission did not clearly present the KLDiv and TVDist metrics. “How well the explanations improve behavior prediction on new inputs” is indeed the definition of “simulatability” that we use throughout the paper. The key idea is that we are interested in distributions of behavior; we define “behavior” as the yes-no probability distribution that the language model gives to the scenario. We also have the predictor give a yes-no probability distribution as its prediction. To measure how good the predictor is, we therefore used the distribution distances KLDiv and TVDist.
>
> Past work has considered binarized model output (taking just the most likely answer, either “yes” or “no”). This allows measuring simulatability with accuracy / F1 score. However, binarizing the model output is throwing away information. We choose to keep the entire distribution to create a more challenging benchmark, where the predictor must distinguish between models which answer “yes” 51% of the time versus models which answer “yes” 99% of the time.
>
> **We have added a discussion of why we use distribution distances to Section 2.3: Evaluation Metrics.**
>
> **In Appendix B, we also added a third quantitative metric, Spearman’s rank correlation coefficient**. We calculate the Spearman correlation between the predictor’s probability of “yes” and the language model’s probability of “yes” across all questions in a topic. This provides another way, different from distribution distance, to measure simulatability. In Appendix B, we see that Spearman correlation results are consistent with the KLDiv and TVDist results: no explanation method does substantially better than the explanation-free control.
>
> > GPT-4's predictions might be independent of input explanations
>
> It’s important to know if GPT-4 has the ability to understand input explanations and change its predictions accordingly. To test this, in Section 4, we provide GPT-4 with explanations for a synthetic linear model. The explanations are hand-crafted to contain prediction-relevant information. **In Figure 4, we see that providing GPT-4 with the “qualitative” and “weights” explanations improves performance compared to the “NoExpl” no-explanation control.** Therefore, we have examples where we know that GPT-4’s predictions depend on the input explanations.
>
> > Have [you] considered more recent sentence embeddings, such as SimCSE?
>
> **We have added experiments with more recent sentence embeddings to Appendix I, including SimCSE and SentenceBERT.** With these new embeddings, we find that the baseline methods have improved performance. Therefore, **we have updated the main results tables (Tables 1, 2, and 3) to use the more recent embeddings.** Our overall pattern of results remains the same: even with new embeddings, no explanation method provides a substantial benefit over the no-explanation control.
>
> > PredictAverage outperforms NoExpl in Figure 4 but does not in Table 1. Could this [discrepancy] be clarified?
>
> Thanks for bringing this to our attention.
>
> PredictAverage depends only on the model being explained (predicting the average probability over the dataset). NoExpl, on the other hand, is GPT-4’s prediction when prompted with few-shot examples of the model’s behavior. We interpret this result as indicating that GPT-4 is better at in-context learning of other language models’ behavior than in-context learning of a synthetic linear model. **We have added this observation to the paper in the “No-explanation predictions” paragraph of Section 5: Results.**

---

> ### Author Response · Authors · 2023-11-20
> **Summary of our Author Response to Reviewer 3Mpe**
>
> Thank you for your review. If we understand correctly, a primary concern of yours is how KLDiv and TVDist relate to simulatability. You also would like to see a deeper analysis of why explanation methods don't improve performance and if GPT-4's behavior is independent of explanations. Finally, you would like more modern sentence embeddings, such as SimCSE.
>
> **We have added additional experiments and analyses to the paper, which we believe addresses all of your concerns.** Please see our other comment for a detailed discussion of these experiments and analyses. In summary:
> - We added to Appendix H an analysis and discussion of possible reasons why the explanations are not improving performance.
> - We have added a discussion of why we use distribution distances to Section 2.3: Evaluation Metrics.
> - We added a third quantitative metric, Spearman’s rank correlation coefficient, to Appendix B.
> - In Section 4, Figure 4, we see an example where the GPT-4 predictor depends on the input explanations to improve its performance.
> - We have added experiments with more recent sentence embeddings to Appendix I, including SimCSE.
> - We have updated the main results tables (Tables 1, 2, and 3) to use more recent embeddings.
> - We have added an observation about the discrepancy between Figure 4 and Table 1 to the “No-explanation predictions” paragraph of Section 5: Results.
>
> _Have we fully addressed your concerns?_ If not, please let us know. We would be glad to incorporate further suggestions into our paper.

---

> > ### Author Response · Authors · 2023-11-22
> >
> > With one day left in the discussion period, we are hoping to hear a reply from you. Please let us know if we have addressed your concerns.

---

> > > ### Comment · Reviewer_3Mpe · 2023-11-23
> > > **Response to Authors**
> > >
> > > > It’s important to know if GPT-4 has the ability to
> > >
> > > It doesn't convince me because the results in Table 1 seems giving different conclusion. "NoExpl" has similar or even lower KLDiv scores compared to other explanation methods. I still challenge that GPT-4 itself can somehow generate good explanations internally and minimize the divergence between various explanation methods.
> > > Without evidence that GPT-4 is able to disentangle the explanation for evaluation, GPT-4 with simulatability is not a good evaluation approach.

---

### Official Review · Reviewer_P4RQ · 2023-11-03

**Soundness:** 3 good
**Presentation:** 3 good
**Contribution:** 3 good
**Rating:** 6
**Confidence:** 3

**Summary:**

The paper presents ALMANACS, a language model explainability benchmark that measures the efficacy of different explanation methods. The benchmark focuses on simulatability, which evaluates how well explanations improve behavior prediction on new inputs. ALMANACS consists of twelve safety-relevant topics with idiosyncratic premises and a train-test distributional shift. The authors evaluate counterfactual, rationalization, and salience-based explanations using another language model as a predictor. The results show that, on average, no explanation method outperforms the explanation-free control, highlighting the challenge of developing explanations that aid simulatability.

**Strengths:**

- The paper addresses the need for a consistent evaluation standard for language model explainability methods.
- ALMANACS provides a benchmark that measures simulatability, a necessary condition for faithful and complete explanations.
- The benchmark includes safety-relevant topics and a train-test distributional shift to encourage faithful explanations.
- The use of another language model as a predictor enables fully automated evaluation, speeding up the interpretability algorithm development cycle.
- The paper presents results that highlight the limitations of current explanation methods and the open challenge of generating explanations that aid prediction.

**Weaknesses:**

- The paper only evaluates the explanation methods based on Kullback-Leibler divergence (KLDIV) and total variation distance (TVDIST). While these metrics provide insights into the performance of the methods, they may not capture all aspects of explanation quality.
- The paper acknowledges that the automated evaluation using language models may not be consistent with human evaluation. Human studies are still needed to validate the results and determine if humans can succeed where language models fail.
- The paper evaluates only three explanation methods (counterfactual, rationalization, and salience-based). While these methods are commonly used in explainability research, there may be other methods that could be valuable to include in the benchmark.

**Questions:**

None

---

> ### Author Response · Authors · 2023-11-20
> **Author Response to Reviewer P4RQ**
>
> Thank you for your review. If we understand correctly, your concerns are that (i) we only use two evaluation metrics, (ii) we do not do a human study, and (iii) we only consider three explanation methods. While we acknowledge the lack of a human study, we have incorporated all your other suggested changes into the paper. In particular, we have **added a third evaluation metric, Spearman's rank correlation coefficient, to the paper**. We have also **added a fourth explanation method, Integrated Gradients**.
>
> _Does this fully address the reviewer's concerns?_ If not, please let us know, and we will be happy to make further changes.
>
> > The paper only evaluates the explanation methods based on Kullback-Leibler divergence (KLDIV) and total variation distance (TVDIST).
>
> In Appendix B, we have added a third metric, Spearman’s rank correlation coefficient. We calculate the Spearman correlation between the predictor’s probability of “yes” and the language model’s probability of “yes.” This provides another way, different from distribution distance, to measure simulatability. In Appendix B, we see that Spearman correlation results are consistent with the KLDiv and TVDist results: no explanation method does substantially better than the explanation-free control.
>
> > automated evaluation using language models may not be consistent with human evaluation
>
> This is an important point; human studies are a valuable direction for future work. Nevertheless, even if an automated evaluation isn’t totally consistent with a human study, we still think automated evaluation has a valuable role to play. Because automated evaluation is an order of magnitude faster than human evaluation, it can provide quick, coarse-grained feedback throughout the interpretability researcher’s development cycle. A tool like ALMANACS is the difference between waiting weeks or months for the results of a human study, versus getting results of an automated benchmark in just a few hours.
>
> > The paper evaluates only three explanation methods
>
> We have added an additional explanation method, Integrated Gradients. Integrated Gradients is a feature attribution method that is axiomatically motivated; it satisfies sensitivity and implementation invariance, and it is the unique path method that is symmetry preserving. (For details, see “Axiomatic Attribution for Deep Networks” by Sundararajan et al.) Integrated gradients was also one of the best-performing methods in Pruthi et al.’s explanation evaluation paper, “Evaluating Explanations: How much do explanations from the teacher aid students?”
>
> In Table 1, we see that the Integrated Gradients results are consistent with the results for other explanation methods: Integrated Gradients does not improve simulatability over the no-explanation control.

---

> ### Author Response · Authors · 2023-11-22
>
> With one day left in the discussion period, we are hoping to hear a reply from you. Please let us know if we have addressed your concerns.

---

### Official Review · Reviewer_Lu72 · 2023-11-09

**Soundness:** 4 excellent
**Presentation:** 4 excellent
**Contribution:** 3 good
**Rating:** 8
**Confidence:** 4

**Summary:**

Simulatability refers to (a human’s) capability to predict model behavior on unseen outputs. Improving simulatability has been considered an important goal for interpretability methods. This paper introduces a new benchmark to automatically evaluate simulatability for interpretability methods, using GPT-4 as a stand-in for humans. Notably, this new benchmark focuses on non-objective tasks with safety-relevant questions.

**Strengths:**

This paper is well-written and easy to follow. By focusing on safety-relevant, non-objective questions, the benchmark differentiates itself well from existing work on interpretability evaluations. The focus on distribution shift also makes the evaluation more realistic than some of the existing work. Overall the paper presents a well-executed idea with very clear motivation.

**Weaknesses:**

As the authors acknowledged, the use of GPT-4 as a stand-in for human annotators limits how much we can take away from the evaluation. Although the paper frames ALMANACS as a benchmark, I find it more suitable to call it a dataset—only when paired with a good-enough human approximator like GPT-4 would it become a benchmark. The lack of user study makes it difficult to judge the evaluation results conducted with the new dataset. But I think the dataset is an interesting starting point for future user studies.

**Questions:**

Given the predictor is GPT-4, it seems like the benchmark can be applied to interpretability goals beyond simulatability. Any thoughts on that?

---

> ### Author Response · Authors · 2023-11-20
> **Author Response to Reviewer Lu72**
>
> Thank you for your comments; our reply is below. **If there is anything else we can do to improve the paper, please let us know. We will be happy to incorporate further suggestions into our work.**
>
> > Although the paper frames ALMANACS as a benchmark, I find it more suitable to call it a dataset—only when paired with a good-enough human approximator like GPT-4 would it become a benchmark.
>
> We agree that the dataset is a primary contribution of ALMANACS. The ALMANACS code we release also provides a complete implementation for using GPT-4 as a predictor; the user needs only to provide an OpenAI API key.
>
> > Given the predictor is GPT-4, it seems like the benchmark can be applied to interpretability goals beyond simulatability. Any thoughts on that?
>
> Yes, we think it’s exciting to consider how ALMANACS can be applied to interpretability goals beyond simulatability. Here are a few possibilities:
> - Studying _minimality_, the absence of extraneous information. To do this, one could randomly sample a subset of each explanation. Then, one could see if the GPT-4 predictor has the same level of performance when looking only at explanation subsets. If a subset of an explanation performs as well as the full explanation, then it suggests that the explanation contains extraneous information. (Although one has to be careful, because having _redundant_ information could be desirable; future work might need to distinguish between _redundant_ versus _irrelevant_ information.)
> - GPT-4 could automate _counterfactual analyses_, like Chen et al.’s work “Do Model’s Explain Themselves? Counterfactual Simulatability of Natural Language Explanations.” Future work could try performing similar sorts of counterfactual analyses in ALMANACS.
> - In order to scale to larger and larger neural networks, we will need _automated interpretability tools_. Because ALMANACS allows fully automated analysis, it can help test scalability of interpretability methods.

---

### Author Response · Authors · 2023-11-21
**Author Response**

Thank you to the reviewers for all their helpful feedback about the paper. We are glad to see reviewers consider our work a "well-executed idea with very clear motivation" (Lu72); "addresses the need for a consistent evaluation standard for language model explainability methods" (P4RQ, 3Mpe, and vWZJ); and appreciate many of our design choices, such as automated evaluation and distribution shift (Lu72, P4RQ, and vQZJ).

We understand the reviewers' primary concerns to be a limited number of language models, evaluation metrics, and explanation methods (P4RQ and vWZJ); how our evaluation metrics relate to simulatability (3Mpe), and lack of a human study (Lu72, P4RQ, vWZJ). While we leave human studies as an important direction for future work, we try to address all other concerns with the following changes:
1. We add experiments with two additional language models for evaluation, in Appendix J. (for vWZJ)
2. We add scaling experiments with nine additional language models, in Appendix G. (for vWZJ)
3. We add a third evaluation metric, Spearman's rank correlation coefficient, in Appendix B. (for P4RQ and 3Mpe)
4. We add a fourth explanation method, Integrated Gradients, to all our main results tables. (for P4RQ)
5. We add a qualitative analysis of possible reasons why explanation methods aren't improving performance, in Appendix H. (for 3Mpe)
6. We add discussion of why we use distribution distance metrics to Section 2.3. (for 3Mpe)
7. We add experiments comparing more sentence embeddings to Appendix I, including SimCSE and SentenceBERT. (for 3Mpe)
8. We update all of the main results tables to use the best-performing recent sentence embeddings. (for 3Mpe)
9. We add discussion of a difference between the experiments of Figure 4 and Table 1, to Section 5. (for 3Mpe)

We provide more details about all of these changes in our individual responses to each reviewer.

**Do these changes address the reviewers' concerns?** Please let us know if concerns remain, and we will be glad to address them.

---

### Meta-Review · Area_Chair_Z1wG · 2023-12-08

**Metareview:**

This paper introduces a new dataset of 12 safety-relevant topics for evaluating explainability methods on natural language. In particular, the paper focuses on “simulatability”, the extent to which the explanation helps a human predicting model’s behavior on novel inputs. The paper provides preliminary results, comparing several commonly used explanation methods with respect to “stimulatibility”, using GPT-4 as a replacement for humans.

Reviewers have raised concerns about whether the measure of stimulability via GPT-4 is reliable at all, the lack of diversity in the evaluation metrics used, explanation methods, models, and discussion about why explanations do (not) help with stimulatibility in the presented experimental setups. Authors have made improvements along some of these dimensions by adding a spearman’s rank correlation coefficient as an additional metric, and integrated gradient as an additional explanation method. They have also added an analysis with more models, comparing logistics regression to accuracy on commonsense ethics which does not seem to directly address the questions raised. They have also added a few qualitative examples and interpretations, which is a starting point to discuss the “why” questions raised.

The paper is a step in the right direction, and the changes made over the discussion period have improved it. But as argued by reviewers, the dataset is not comprehensive enough to be considered as a benchmark. Introducing a new dataset, especially one to be considered for benchmarking, needs more in-depth discussion and analysis. Presenting a hold-out set is not enough to claim addressing needs for “distributinal shifts”, for example. What kind of distributional shift? Topic? Length? Language? Phrasing? Please also consider adding a datasheet.

Perhaps the most pressing concern which is not addressed at all is the use of GPT-4 as a replacement for humans. Authors have referred to their preliminary study showing hand crafted explanations in a constrained synthetic setup can help with “stimulability” of GPT-4. But this is not a strong enough piece of evidence to use it as replacement for humans, without any human-subject study. In addition, the gap between this preliminary study and the main experiments is too large, both in terms of the choice of tasks and explanations.

For these reasons, this paper can benefit from another round of revision to better motivate the choice of GPT-4, provide a more comprehensive discussion and interpretation of the observations. In addition, I’d like to add that examples are indeed a form of explanation and I encourage the authors to revisit their presentation of noExp setup and perhaps add another baseline condition.

**Justification For Why Not Higher Score:**

- The paper claims to present a new benchmark, but the dataset is not comprehensive enough and the design choices are minimally discussed.
- The choice of GPT-4 as an oracle for stimulatibility needs more evidence. There is a big leap between the preliminary evidence, and the main experiments.
- Based on AC’s read, the baseline choice is indeed another form of explanation through examples, which makes the current presentation of findings potentially misleading.

**Justification For Why Not Lower Score:**

N/A

---

### Decision · Program_Chairs · 2024-01-16

Reject